



# Brief communication: A continuous formulation of microwave scattering from fresh snow to bubbly ice from first principles

Ghislain Picard[1,2], Henning Löwe[3], and Christian Mätzler[4]

[1]Univ. Grenoble Alpes, CNRS, Institut des Géosciences de l'Environnement (IGE), UMR 5001, Grenoble, France
[2]Geological Survey of Denmark and Greenland (GEUS), 1350 Copenhagen, Denmark
[3]WSL Institute for Snow and Avalanche Research SLF, Davos, Switzerland
[4]GAMMA Remote Sensing AG, Gümligen, Switzerland

**Correspondence:** Ghislain Picard (ghislain.picard@univ-grenoble-alpes.fr)

**Abstract.** Microwave remote sensing of the cryosphere demands a formulation of the scattering coefficient which can be applied over the entire range of relevant densities, from fresh snow to bubbly ice, at all frequencies and for any grain size and snow type. Most challenging are intermediate densities (450–550 $\mathrm{kg m^{-3}}$) and high frequencies (or coarse-grained snow) where current scattering formulations break down. In this brief communication we demonstrate that the strong contrast expansion

method, recently developed for heterogeneous, dielectric media can be applied to microwave scattering in snow, firn and ice for solving these problems.

## 1 Introduction

Optimal use of satellite observations to retrieve information from the snowpack requires a precise understanding of the interaction between electromagnetic waves and snow. In the microwave domain, a thorough model for the radiation emitted or

reflected by a snowpack involves three main ingredients: 1) A snow microstructure representation, which determines the input parameters that must be collected in the field or obtained from a snowpack evolution model; 2) An electromagnetic theory to compute the scattering and absorption coefficients in each snow layer from the given microstructure parameters; 3) A method to solve the radiative transfer equation that describes the propagation of the radiation from layer to layer up to the surface. Well established and accurate methods exist for the last ingredient as long as the snowpack has a plane-parallel layered structure

(e.g. Tsang et al., 2000; Jin, 1994). The first ingredient is an active research topic that has received much attention in the last decades(e.g. Mätzler, 2002; Royer et al., 2017; Sandells et al., 2021) because it is a major source of uncertainties. The second ingredient is the topic of this communication.

Established rigorous and empirical electromagnetic theories have been used for decades by the snow community. The most popular ones are the HUT empirical relationship (Pulliainen et al., 1999), the IBA approximation (Mätzler, 1998) and the

DMRT QCA theory (Tsang et al., 1985). Despite being used with reasonable success, all of these formulations have some fundamental, and therefore practical, limitations. These limitations originate from restricting assumptions in these theories about the grain size, grain shape and snow density which prevent consistent microwave modeling of snow, firn and ice in the





entire range of density and microstructures found throughout the cryospheric regions. The development of robust retrieval algorithms of snow properties requires improved consistent scattering theories.

In what follows, we shall pinpoint these limitations and their implications and demonstrate how they can be simultaneously overcome by employing new, theoretical results on the effective permittivity of random porous media. In a recent paper (hereinafter TK21, Torquato and Kim, 2021) propose the non-local, strong contrast expansion (SCE), as a generic homogenization theory to compute effective wave propagation in random two-phase media. A careful inspection of the comprehensive (54 pages) derivation and of two precursor papers (Rechtsman and Torquato, 2008; Kim and Torquato, 2020) reveals that it

sheds new light on three key problems concerning the grain size, grain shape and density. The objective of this communication is to relate this new theory to existing ones and to feature the powerful approach (Torquato and Kim, 2021) through its implementation in the Snow Microwave Radiative Transfer model (SMRT, Picard et al., 2018).

## 2    The strong contrast expansion

Our starting point is the main result of the strong contrast expansion (Rechtsman and Torquato, 2008; Kim and Torquato, 2020;

Torquato and Kim, 2021) which expresses the dielectric polarizability of the effective snow medium (in any dimension $d$) with respect to the background (air) $\beta_{\text{eff-air}}$ as an *exact* infinite series in the dielectric polarizability of ice with respect to air $\beta_{\text{ice-air}}$. Here, we consider the case of an isotropic 3-dimensional medium composed of ice and air. The effective polarizability reads (eq. 54 in TK21):

$$\beta_{\text{eff-air}}^{-1} \phi_{\text{ice}} \beta_{\text{ice-air}} = 1 - \frac{1}{\phi_{\text{ice}}} \sum_{n=2}^{\infty} \beta_{\text{ice-air}}^{n-2} A_n^{(\text{ice})}(k_{\text{eff0}}) \tag{1}$$

where $\phi_{\text{ice}}$ is the fractional volume of ice and $k_{\text{eff0}}$ the wave number in the effective medium in the static approximation (this version is referred as to "scaled" SCE in TK21 section VI.B.2 and is used here for consistency with the other theories). The polarizability for any medium or material 1 and 2 is defined as:

$$\beta_{\text{1-2}} = \frac{\epsilon_1 - \epsilon_2}{\epsilon_1 + 2\epsilon_2} \tag{2}$$

To simulate the snow electromagnetic properties, a model shall first deduce the effective permittivity $\epsilon_{\text{eff}}$ of the ice+air medium

using Eq. 2 from $\beta_{\text{eff-air}}^{-1}$ calculated with Eq. 1, once the other terms are calculated. The extinction coefficient is deduced from the imaginary part of the effective dielectric constant $K_e = 2k_0 \Im \epsilon_{\text{eff}}$ where $k_0$ is the wave number in the vacuum. The absorption coefficient is obtained in the static regime ($k_0 = 0$ or formally by setting $A_n = 0$ for all $n$). The scattering coefficient is finally obtained by $K_s = K_e - K_a$.

    The terms $A_n^{(\text{ice})}$ are complicated expressions (eq. 51–53 in TK21) but only depend on the microstructure and the wave

number. They carry all the geometrical information of the medium microstructure (only the fractional volume appears in other parts of the theory) and are thereby the main driver of the scattering signature of the medium. $A_2^{(\text{ice})}$ is an integral over space of the two-point correlation function. Roughly speaking, this function accounts for the size, shape and relative arrangement in space of the ice crystals. It can be obtained by computing the auto-covariance of 3D images of the snow (Sandells et al., 2021).





This $A_2^{\text{(ice)}}$ term is similar to the $I$ integral in IBA (Mätzler, 1998) and is linked to the pair-correlation in DMRT (Tsang and
Kong, 2001; Löwe and Picard, 2015).

The term $A_n^{\text{(ice)}}$ depends on all $n$-point correlation functions up to order $n$. This implies that the calculation of the terms
$n > 2$ would require increasingly more detailed information of the microstructure, which is hitherto not available for snow in
practice (Sandells et al., 2021). Not mentioning the difficulty of the numerical computation of these complex terms, we thereby
consider the term $n = 2$ only at this stage. The numerical implementation of SCE and of the other approximations presented
below have been done in the open source SMRT model (Picard et al., 2018).

## 3 Assumptions on size

As stated in their title, TK21 develops the "non-local" SCE extending former work on the "local" SCE (RT08). The latter
requires very small sized scatterers with respect to the wavelength (i.e. the quasi static approximation) and the absence of large
aggregates of scatterers, which makes it more suitable to low microwave frequencies (the equivalent terminology in DMRT
is long and short range approximations Tsang and Kong (2001) which was adopted in SMRT). The main difference between
the two versions appears in the $A_2^{\text{(ice)}}$ term and implies a numerical integration accounting for the surrounding microstructure
instead of a simple function evaluation. The extra computational cost and risk of numerical instability is to be balanced with
the gain in accuracy. To assess this gain for snow, Fig. 1 presents the scattering coefficient for sticky spheres of radius $a$ as a
function of $k_0 a$ (the simulations are run with a fixed radius of $0.5\,\text{mm}$, stickiness 0.2 and varying frequency up to $150\,\text{GHz}$).
The results for a density of $300\,\text{kgm}^{-3}$ clearly show that the short range theories (RT08 and the two DMRT flavors available
in SMRT) diverge around $k_0 a \approx 0.6$ from the long range theories (TK21, IBA, and the Mie DMRT version implemented in
the DMRT-QMS model Tsang et al. (2007)). The long range theories are close to each up to $k_0 a \approx 1.5$ (note that the lower
scattering coefficient with SCE compared to IBA and Mie DMRT is a result of the theory, not a convergence issue) and they
diverge for even larger grain sizes. In principle, SCE has the capability to work in this extended range provided the terms $A_n^{\text{(ice)}}$
with $n > 2$ are calculated. However, TK21 also indicates a fundamental hard upper bound of the order of the wavelength (i.e.
$k_0 a \lessgtr 6$).

Overall, these results give additional confidence in the three long range theories, even with the $n = 2$ truncation, to be
applicable for most snows in the microwave range. Moreover TK21 conducted precise electromagnetic simulations (FDTD
technique) that compare favorably with SCE for a dielectric contrast comparable to that of ice and air, and up to $ka \approx 1$ (fig. 9
in TK21). These results also highlight the equivalence between SCE, IBA and DMRT Mie. Our main practical recommendation
is limiting the size to $k_0 a \lessgtr 1.5$ in any simulation.

## 4 Assumption on shape

The different scattering theories require some assumptions on how the medium microstructure is prescribed and these assump-
tions have a critical impact on the scattering signature. In the original DMRT (Tsang et al., 1985; Tsang and Kong, 2001)



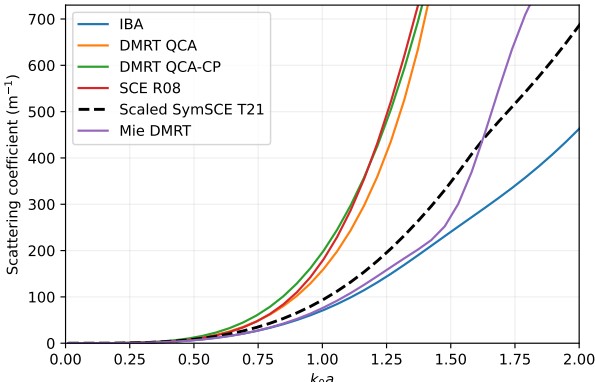

**Figure 1.** Scattering coefficient as a function of the product of wave number and radius ($k_0 a$) for a variety of electromagnetic theories. The microstructure is made of sticky hard spheres with radius $0.5\,\mathrm{mm}$, stickiness 0.2 and density $300\,\mathrm{kg\,m^{-1}}$.

spherical particles are assumed with their relative positions determined by the sticky hard sphere model. Despite the success of this theory, snow is composed of diverse shapes interconnected in a complex manner which prevents a successful mapping on the sticky hard sphere model (Löwe and Picard, 2015). In the IBA, this restrictive assumption on the shape of the scatterers is partially removed. While IBA still assumes a particular shape to compute the internal field ratio in the scatterers, it lets free the choice of the two-point correlation function which can be accurately constrained by data (Sandells et al., 2021). However technically it is possible (and practically done) to select a shape and a form of correlation function independently and inconsistently (e.g. spherical scatterers and an exponential correlation function). The internal field ratio can also be optimized from experimental data (Mätzler, 1996, 1998) without explicitly assuming a geometrical shape.

The SCE derivation does not rely on the scatterer concept, the medium is described by the point correlation functions and the electromagnetic derivation uses the Green function formalism (called bi-local approximation in Tsang and Kong, 2001). Nevertheless, an infinitesimal exclusion volume is required to integrate the Green function, and its shape determines the form the SCE final equations. TK21 choose a sphere for this volume (eq. 25 in TK21) in the main text and illustrate the results for two alternative shapes (appendix A in TK21, eqs A1 and A2). These three expansions constitute different forms but are strictly equal in the infinite series. However, in practice, the necessary truncation breaks this equality. Further noting that the spherical exclusion volume provides a faster convergence (TK21 sec III B) as a function of the dielectric contrast, the conclusion is that a spherical volume is recommended, independently of the actual microstructure, i.e. independently of the real shape of the scatterers.

Interestingly, by analyzing the similarities between IBA and SCE expressions, it appears that choosing the scatterer shape in IBA and of the exclusion volume in SCE lead to similar analytical polarizability equations. The numerical results also highlight the similarity between both theories (Fig. 2). We conclude that while getting rid of the scatterer concept is a theoretical advance of SCE, it does not lead to numerical improvements in practice.





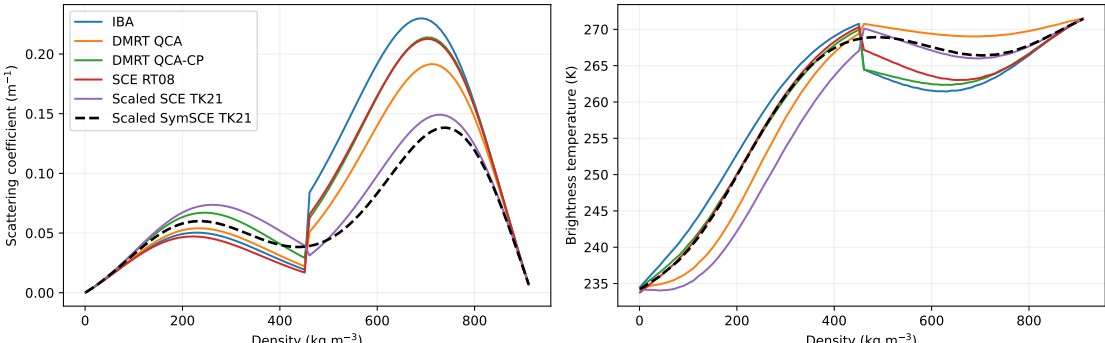

**Figure 2.** Scattering coefficient as a function of density for a variety of electromagnetic theories. The microstructure is made of sticky hard spheres with radius $0.2\,\mathrm{mm}$ and stickiness 0.2. The frequency is $19\,\mathrm{GHz}$.

## 5 Assumptions on density

Despite being developed for dense media, the existing theories used for snow actually become inaccurate at high density. The DMRT theory was shown to degrade for fractional volume $\phi_{\mathrm{ice}} > 0.3$ (Liang et al., 2006) when compared to exact electromagnetic calculations, possibly because of the limitation of the Percus-Yevick approximation applied for solving the sticky hard sphere model. The SCE theory is invalid when the ice phase is percolating (RT08) which also occurs from $\phi_{\mathrm{ice}}$ around 0.3 for many microstructures. Regarding IBA, the apparent permittivity is valid for $\phi_{\mathrm{ice}} \lesssim 0.5$ and little is known about the other parts of this theory. This general limitation is severely restrictive for application to snow where density is often larger than $300\,\mathrm{kgm^{-3}}$ especially on glaciers and in the firn.

For very large density, when the fractional volume of air is $< 0.3$, a possible workaround is to "inverse" ice and air in the equations. No theoretical limitation forbids this inversion. In any case, a wide range of common intermediate densities remains inaccessible ($\approx$300-700 $\mathrm{kgm^{-3}}$). RT08 and Dierking et al. (2012) independently suggest the following method: the theory is applied up to the percolation limit on both sides ($\phi_{\mathrm{ice}}$ =0–0.3 and 0.7–1) and spline interpolation is used in between. This ad hoc correction is effective to smooth the discontinuity that appears when using medium inversion at $\phi_{\mathrm{ice}} = 0.5$ (Fig. 2) but it lacks of physical ground.

An alternative rigorous approach is proposed in TK21. Considering that i) the infinite expansion Eq. 1 is exact, and that ii) the same expansion applied with ice and air switched is exact as well, then, any combination of both expansions is exact as well. Let choose a linear combination weighted by $1 - \phi_{\mathrm{ice}}$ and $\phi_{\mathrm{ice}}$ for the normal and inverted medium respectively, so that a higher weight is given to the normal medium at low $\phi_{\mathrm{ice}}$ and to the inverted medium at high $\phi_{\mathrm{ice}}$. Once truncated at $n = 2$, this combination yields a quadratic equation in $\epsilon_{\mathrm{eff}}$ that is solved analytically (eq. D2 in TK21). This provides a new approximation of SCE that we call SymSCE.

Fig. 2 shows the scattering coefficient computed for several theories and approximations as a function of snow density. Sticky hard spheres with radius of $0.3\,\mathrm{mm}$ and stickiness of 0.2 are used for all the simulations. For very low and very high





ice fractional volumes, all theories give similar results showing an excellent agreement between the new SymSCE theory and the established ones. They however all differ in the intermediate range ($150 - 800\,\mathrm{kgm}^{-3}$, often moderately, but by a factor of up to 2 around $\phi_{\mathrm{ice}} = 0.5$ ($458\,\mathrm{kgm}^{-3}$). For comparison, this value can be compared with the 1.5-fold uncertainty in scattering resulting from the typical 15% uncertainty on measured optical radius of snow (Gallet et al., 2009). As expected all but SymSCE are affected by a discontinuity around $\phi_{\mathrm{ice}} = 0.5$ where we chose to invert the materials. Despite this nice and improved behavior, it still remains to prove that SymSCE is more accurate than the other theories. Fig. 2 also shows the consequences on brightness temperature for an hypothetical semi-infinite single homogeneous layer of snow (at 273K). The overall trend is an increasing function, mainly driven by the monotonous increase of the absorption with density. Scattering is modulating this overall trend. Despite this secondary role, the differences between the theories reach $10\,\mathrm{K}$ in a wide range of densities, a significant uncertainty.

SymSCE is not the unique possible combination of SCE equations to yield a symmetrical behavior (e.g. $1 - \phi_{\mathrm{ice}}^2$ and $\phi_{\mathrm{ice}}^2$ is possible). However, there is one reason to prefer this particular definition. In the static regime, only the SymSCE reduces to the Polder and van Santen mixing formula (Polder and van Santen, 1946), which is the most accurate theoretical equation to predict the real part of snow permittivity (Mätzler, 1996; Olmi et al., 2021). It is worth noting that the procedure of linear combination is not specific to SCE, it was previously proposed to symmetrize the Maxwell-Garnett mixing formula, yielding the Polder and van Santen mixing formula (Sihvola, 1999) and it could be applied on IBA and DMRT equations to remove the discontinuity.

## 6  Conclusions

The non-local symmetrized SCE theory presented in TK21 and featured in this brief communication provides several conceptual remedies to long-standing problems in other theories commonly used to compute snow scattering from microstructure information. The key point is that the strong contrast expansion is exact and does not require any assumption until the truncation at the end of the development. The actual accuracy of this theory for practical snow simulations is however not known and could only be assessed with detailed ground truth after eliminating other major uncertainties like the snow microstructure, soil parameters, surface roughness, etc.

Nevertheless, the SCE as presented in TK21 is even more general than applied here, paving the way to further improvements. For instance, the calculations of the higher terms of the series ($n > 2$) should extend the validity range to higher frequencies and/or coarser-grained snow, provided that more detailed microstructure information could be obtained. The general case of anisotropic media is explicitly treated in TK21, which is needed to explain some experimental results on snow scattering at low frequencies (Leinss et al., 2016). At last, we suggest that further work should assess the accuracy of the different long range theories which will require more precise in-situ microwave and snow measurements than available at present.





*Code availability.* The SMRT model code is available from https://github.com/smrt-model/smrt. The SCE implementation will be added in this repository once this paper is validated by the reviewers. The code to produce the figures will be posted as well https://github.com/

smrt-model/smrt_sce_paper. The DMRT-QMS model is available from https://web.eecs.umich.edu/~leutsang/ComputerCodesandSimulations.html.

*Author contributions.* GP conducted the study, implemented the SCE in SMRT and run the simulations. HL and CM contributed to the analysis and the results. All authors contributed to the manuscript.

*Competing interests.* The authors declare not competing interests

*Acknowledgements.* This work was funded by ESA Project 4000112698/14/NL/LvH "Microstructural Origin of Electromagnetic Signatures in Microwave Remote Sensing of Snow".





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
