# Peer review of "Brief communication: A continuous formulation of microwave scattering from fresh snow to bubbly ice from first principles"

_The Cryosphere, 2022_

## Author Response (AR1)

The authors would like to thank the reviewer for their comments and feedback. Our responses are presented in blue.

**Reviewer 1**

The paper applies the formulation developed by TK 21 (Torquato and Kim, Physical Review X 2021) to scattering by fresh snow. The merit of the non-local formulation is to extend the previous quasistatic model (Torquato 2002) to higher frequency so that the attenuation due to scattering is accounted for. The attenuation in scattering is included in the imaginary part of the effective permittivity. The TK21 method is supposedly valid up to ka =6 where k is the wavenumber and a is grain radius. The value ka=6 means the gran radius is 1 wavelength

There are questions whether the TK21 model is applicable to snow

a.    A microscopic picture of snow shows that the ice granis have irregular shapes and that they also have "stickiness" that the grains adhere together.   A limitation of classical mixture formula as given in the book by Sihvola (1999) is that the mixture formula depends on the shape of the scatterers. The mixture formula is developed for simple shapes such as spheres, ellipsoids, disks etc. The problem with irregular shape is that the solutions of Maxwell equations are "discontinuous "across the boundaries between the scatterers and the background. Boundaries conditions indicate that although the tangential component of electric field is continuous, but the normal component electric field is discontinuous with the normal component in air that is 3.2 times that of in ice.  For well-defined shapes such as ellipsoids, such boundary value problems are solved by ellipsoid coordinates. But such problems cannot be solved analytically for irregular shapes The TK21 model is strongly dependent on the choice of exclusion volume which is analogous to particle shape.  The examples in the paper TK21 are limited to regular shapes such as spheres, disks etc.  In TK21, the beta pq in equation (33) with d=3 indicates the shape of a sphere.

This comment raises 2 main points, **the boundary problem** and the question of the equivalence between the **exclusion volume and the particle shape**.

It concerns our section 4 which aims to highlight that TK21 theory does not need the notion of individual particles (with some shapes and sizes) even though it solves (exactly up to some point) the Maxwell equations and hence, takes the boundary conditions into account, overcoming the problem highlighted by the reviewer.

More precisely, regarding **the boundary problem**: We agree that classical formulations explicitly solve the boundary problem for given particles, which is relatively straightforward for simple geometrical shapes, but is not for irregular shapes. TK21 does not solve this problem explicitly but implicitly because the medium is not represented by particles but by local information on whether phase 1 or 2  (air and ice in our case) is present at any point r in space. Actually, this information is

given in a statistical way by the n-point correlation functions. For a medium made of particles, the n-point correlation functions hold all the information about both the shape and the relative arrangement of the particles. However TK21 approach is more general because it also applies to non-granular media, when particle does not exist (or a inter-connected or overlaps). In this sense, the TK21 formalism is "non-classical", it has a much wider range of applicability in ters of type of media than previous classical formulas. TK21 is not unique in this category, IBA and the Strong Fluctuation Theory (SFT) share similarities about the medium description and how the electromagnetic equations are derived.

Regarding **the exclusion volume and its shape**: While it is correct to state that TK21 formalism results in different formulas depending on the shape of the exclusion region associated with the Green's function singularity, this volume is not analogous to the particle shape. The notion of particle is not used anywhere in the derivation of the equations. This exclusion region is infinitesimally small and is not a physical entity. It is a mathematical entity. For example, choosing a spherically-shaped exclusion region does not at all mean one can only apply such formulas to spherical particles. One can still apply them to non-spherical shapes, including cubes, ellipsoids, etc. Importantly, as alluded above, the n-point correlation functions incorporate the information of particle shapes, in an indirect manner. Despite this fundamental difference, the fact that the equation (33) in TK21 corresponds to that of spherical particles is certainly not a coincidence, it is because the polarizability of a spherical particle (smaller than the wavelength) is the same as the polarizability of a spherical exclusion volume, since it does not depend on the size (as long as the spheres are small). But the similitude stops there.

TK21 goes a step further by indicating that rather than being related to the particle shape, the shape of the exclusion region is actually related to the symmetry of the effective dielectric constant tensor. For example, a spherical exclusion volume gives an isotropic effective dielectric constant and thus is optimal when the effective dielectric constant tensor is expected to be isotropic. This point is novel to our knowledge and important, this is where TK21 advances from IBA and SFT which relied on some assumptions about the particle shape. Some similar element was already in the Improved Born Approximation via the K field factor proposed in Mätzler, 2002, but TK21 makes it more explicit.

Our text is faithful to this response, and only minor editing is proposed:

"The SCE derivation does not rely on the scatterer concept, the medium is **exhaustively** described by the **n-point** correlation functions and the electromagnetic derivation uses the Green function formalism \citep{tsang_2001_vol3}."

"Further noting that the spherical exclusion volume provides a faster convergence (TK21 sec III B) as a function of the dielectric contras**t in the case of isotropic media**, the conclusion is that **a spherical exclusion volume is recommended for any isotropic microstructure, i.e. independently of the existence (and shape) of individual scatterers.**"

b.    In computational electromagnetics, such boundary value problems of irregular shape/boundary have been handled by more accurate numerical techniques such as edge elements in vectorial finite element method, RWG basis function in the method of moment, and Nystrom method for volume integral equations. The popular FDTD method which is used in TK21 is not accurate This is because

FDTD uses rectangular grids in the discretization. It is unclear that the formulation of TK21 can handle the irregular shape of ice grains to correctly obey the boundary conditions on the surface of a scatterer.

It is not possible to explicitly check that the boundary conditions on the surface of a particle is verified, at least because the description of the medium is statistical in TK21. However the starting point of TK21 theory is the wave equation in a two-phase medium (which is an exact consequence of the Maxwell equations) and the development is very similar to that of other authors (e.g. Strong Fluctuation Theory, Chap 4, Section 3, Tsang and Kong 2021). The key breakpoint is from Eq 38 in TK21 and the benefit of this new treatment is explained in Remark viii. In TK21.

Regarding FDTD, we remove the term "precise" in our paper, which was to be understood relative to the analytical formulations, not to other numerical methods. However, we admit this was misleading.

"Moreover TK21 conducted **numerical** electromagnetic simulations using the Finite-Difference Time-Domain method that compare favorably with SCE for a dielectric contrast comparable to that of ice and air, and up to $k a \approx 1$ (fig. 9 in TK21)"

c.    For TK21, the methodology is based on point geometries and correlation functions associated with point geometries.  In principle, the point geometry has correlation functions of infinite order which makes the solution "exact". But in practice to infinite order is not possible.  Only the second order correlation function is used which means that the solutions have inaccuracies.

We fully agree with this and this point is explicitly addressed (last two paragraphs in Section 2) in the initial version of the paper, where we state our use of the truncation for n=2. This is not ideal as discussed in Section 2 but it is equivalent to the approximations found in the other formulations that we consider in our Section 3  (namely DMRT QCA and IBA). In this respect, TK21 with the truncation n=2, is the state of the art for snow, not beyond. However the TK21 methodology is not restricted to "point geometries" (in the sense of decorated point processes) a similar hierarchy emerges for continuous, non-particulate phases.

A benefit of TK21, for the future, is that the expressions for higher order truncation (n>=3) are given explicitly which paves the way for further work. However, applying the theory for n>=3 is very challenging for two reasons: 1) the n-point correlation function is required, and although a statistical estimator of these functions can be obtained from micro-computed tomography images, the inaccuracy is certainly very high based on our experience (e.g. Loewe and Picard 2015, Sandells et al. 2021). Our intuition at this stage is that the inaccuracy will cancel the benefit of using such advanced truncation (n>=3), but this is very open.

2) the numerical implementation of the terms n>=3 is not trivial, and it is likely that the expressions in the form given by TK21 would be subject to numerical instability. A bit more mathematical work is needed even though TK21 already provides a significant advance.

At last, we question the interest of exploring this route, as of 2022, given that practical field snow measurement techniques (excluding micro-computed tomography,) are actually insufficient to even characterize the 2-point correlation function with full accuracy.

The curren paragraph addressing this question has been slightly amended:

"**For a given order $n$,** the term $A_n^{\textrm{(ice)}}$ depends on all $n$-point correlation functions up to that order. This implies that the calculation of the terms $n>2$ would require increasingly more detailed information of the microstructure **(e.g. larger and more resolved 3D images of the snow microstructure)**, which is hitherto not available for snow in practice \citep{sandells_2021}. Not mentioning the difficulty of the numerical computation of these complex terms, we thereby consider the term $n=2$ only at this stage. "

d.    For ka extended beyond 0.5, there is incoherent field that contributes to radiative transfer equation. In addition to the attenuation due to scattering, there also is the phase matrix.  This part of phase matrix has not been treated in TK21. Recently, the cross-polarization of the phase matrix at C band, X band and Ku band have drawn significant interests in microwave remote sensing of snow.

This is correct, the phase matrix is not explicited in TK21. It could be derived from their equation 39 with probably very significant work. In our paper, most of the results concern the scattering coefficient only. However to produce Fig 2, we had to assume a phase matrix. We did so by taking the same form as predicted by IBA. This information was missing in our first version, it is now stated in the manuscript:

"Note that for this calculation,  the phase matrix of the radiative transfer equation is needed. It is not available for the SCE theory and we assumed that its angular variations are the same as predicted by IBA."

We are confident that this hypothesis is valid because of the proximity between IBA and TK21 at n=2. However, more mathematical work is needed to demonstrate this result.

e.    In Mie scattering, there are two series with two sets of coefficients: one is "electric" and the other is "magnetic". For ka<<1, the electric series dominate. However, when ka gets larger such as in TK21, the magnetic series contribute.  It is unclear that TK21 include the magnetic series if the model is applied to dense Mie scattering

It is unclear which approximation in TK21 motivates this concern. As explained above, the medium description and the derivation of the electromagnetic equations are very different between TK21 and particle-based theories (such as Mie), which makes it difficult to compare the equations.

Although I have the above reservations about the applicability of TK21 to snow, I do think it is worthwhile for this paper to compute the results of attenuation of TK21 in snow and compare with other models.

 I recommend the following revisions of this paper.

1, The authors should discuss the 5-bullet points a, b, c, d, and e that are raised above.

2. In figure 1, the frequency dependence power law should be extracted and tabulated. For Rayleigh scattering, it is frequency to the 4th power. The power law dependence makes the model comparisons easier to digest and remember.

We include here a loglog plot equivalent to Fig 1 with a power law fit. We found (dark gray curve) that SymSCE behaves as a power law of 3.3 around ka=1 and for the specific conditions (density, stickiness) of Fig 1. However this power law is quite different for other ka, density and stickiness which makes this information context-specific. While this figure provides interesting information, the main message in our text is the divergence between the short vs long range theories first, and then between the long range theories, and for this, the actual Fig 1 is clear. Since in addition tabulating exponent values or adding a new figure are not suitable for the short format of this paper, we have briefly integrated this question of the power law in the main text in the paragraph on the interpretation of Fig 1, as follows:

[Figure]

"The results for a density of 300 kg m$^{-3}$ show a sharp increase as a function of $k_0 a$ for all models. This increase indeed follow a power law in $k_0$^4 at low frequencies which tend to slow down in the case of the long range theories (TK21, IBA, and the Mie DMRT version implemented in the DMRT-QMS model \citet{tsang_2007}) when $k_0 a$ is increasing (e.g. we found an apparent $k_0$^3.3 relationship for SymSCE if fitted around $k_0 a$=1)."

3. The Mie-DMRT in figure 1 may not be correct. This is because when ka exceeds 1, more terms in both the electric series and the magnetic series should be included. It is unlikely that Mie-DMRT go off like that as in figure 1. The Mie-DMRT has weaker frequency dependence than the power law of 4. I suggest that the author delete the Mie-DMRT unless their results are correct.

The behavior seems non natural indeed. The Mie DMRT was obtained with the DMRT-QMS model that may involve numerical approximations responsible for this unexpected behavior. We propose to keep the curve (which is the correct results of the numerical calculation) and relabel the legend with

DMRT-QMS instead of Mie DMRT to make clear that the curve is specific to one implementation (this was already explicit in the text) and that the problem may come from the implementation.

We prefer not to remove the "strange" part of the curve because the purpose of this figure is to identify when the different theories diverge from each other. Our belief is that even though IBA and SCE (with n=2) seems to show a more continuous behavior for ka>1.5, it is not sufficient to trust their results. All these three theories are not supposed to work in this high range, and we interpret the divergence noted by the reviewer between these theories as an illustration of this global limitation, not as a specific limitation to one or another theory.

4. The results in figure 2 should be done for larger grain size. At least, a new figure with a=0.4 mm should be added at 19GHz. The use a=0.2mm is too small. The scattering coefficient is only a fraction per meter which is too small for real life problem. The measured volume radar backscattering at 19 Ghz is much larger for a snow depth of 1 meter.

Actually the graph was produced with a=0.3mm as stated in the text (the legend was erroneous), but the value in our legend was erroneous. We have increased the radius a=0.4mm to follow the Reviewer's comment and have updated the figure.

We have also propagated this change in the text, specifically on the value of the difference between the theories which is now a bit larger: "Despite this secondary role, the differences between the theories reach 15\,\unit{K} in a wide range of densities, a significant uncertainty."

5. In equation (1), the summation over index n is up to infinity. However, in practice only second order, n=2, is used. The results in this paper are based on A2. The expression for A2 should be explicitly given so that readers can write a computer code for A2 readily.

To keep the paper in the brief communication format, we prefer not to include the A2 equation which is quite complex and requires the definition of many terms, needing space. But we agree that direct access to the equation is helpful so we have added in our text the precise equation number for A2 in TK21.

Regarding the implementation of such an equation, TK21 makes it clear that the numerical computation is tricky and provides useful hints in supplementary materials (eq S111). We will provide a computer code (in the SMRT model) as open source on github and in the Zenodo permanent repository, as stated in the code availability section. Our code is fully documented with all the equation numbers referring to TK21. Verification and improvement of our implementation by other research groups are facilitated as much as possible.

**Citation**: https://doi.org/10.5194/tc-2022-63-RC1

The authors would like to thank the reviewer for their comments and feedback. Our responses are given below in blue.

**Reviewer 2**

This is a welcome short communication that addresses an important issue. The authors proposed to apply a new method for non-local strong contrast expansion, developed by Torquato and Kim (2021) to compute wave propagation in a two-phase media, to the full range of snow density and microstructure found throughout cryspheric regions. The intention is to help to address problems of grain size, shape, and snow density in electromagnetic theory, particularly related to the computation of the microwave scattering coefficient. The paper is well-written and concise. There are several corrections recommended for clarity, and a suggestion for expansion of the discussion. Several other minor corrections are recommended, listed sequentially.

General comments

- Can the authors comment on the general applicability for active microwave modelling? Or is this applicable only to passive microwave solutions?

The SCE is concerned by the propagation of electromagnetic radiation in porous medium (effective permittivity), whatever the source of this radiation. Thereby, as a general response, it applies to both passive and microwave remote sensing equally. However, since the radar and passive microwave radiometer may operate in different ranges of frequency, the benefit is to be evaluated on a case by case basis, for each sensor and each application.

We have added mention of "passive and active microwave" in the conclusion and in the Section 2 "The numerical implementation of SCE and of the other approximations presented below have been done in SMRT, an open source, **active and passive microwave** radiative transfer model "

- We have often observed enhanced scattering where substantial depth hoar (DH) is present. Does the work of TK12/SCE support/explain this behaviour wrt DH given the distinct microstructure? Given the strong scattering from DH, it would be nice to see some comments on suitability for DH. The authors state in the abstract that this method should be applicable for coarse-grained snow and again in the conclusion but specific reference to DH would be welcome.

As stated in the introduction, it is useful to distinguish three main steps in modeling the propagation of radiation in the snowpack, and this paper is about step 2 (electromagnetic theory) while the reviewer's question is covered by our step 1 (microstructure). We have a new paper published very recently (Picard et al., 2022) which specifically addresses step 1 in detail, and contributes to the answer to the question of DH scattering efficiency.

G. Picard, H. Löwe, F. Domine, L. Arnaud, F. Larue, V. Favier, E. Le Meur, E. Lefebvre, J. Savarino, A. Royer, The Microwave Snow Grain Size: A New Concept to Predict Satellite Observations Over Snow-Covered Regions, *AGU Advances*, 3, 4, https://doi.org/10.1029/2021AV000630

In a nutshell, yes, depth hoar is an efficient scattering crystal because in general depth hoar crystals are bigger in extent than other grains (a well-known trait) and also because it has a specific structure that we demonstrate in picard et al. 2022 to the related to the chord length dispersity which makes it more efficient for a given "size" (this is the main new finding in Picard et al. 2022).

In the present paper, our use of "coarse-grain" is generic and is relative to the frequency. DH can be either small (e.g. at C-band) or big (e.g. Ka band). We have not amended the text because we believe the question of DH scattering is out of the scope of the present paper. However, we add the reference to Picard et al. 2022 which was not accepted at the first submission of the present paper.

Specific comments

- on Line 22, the authors state that existing theories "prevent consistent modelling of snow". Can the authors clarify what they mean by consistent?

We have checked the definition of this term, which is "a theory that does not suffer from assumptions" and it is conform to what we mean, that is a theory that is valid for "snow, firn and ice in the entire range of density and microstructures found throughout the cryospheric regions" (extract of the actual text). Existing theories are not consistent (at least) because representing snow and dense firn is obtained by material inversion, and the mid-range of densities can not be represented.

- Line 21-24. Related to the above point, perhaps the authors can cite specific studies that

The comment is incomplete but we can say that there are no study to our knowledge that demonstrates experimentally the problem in the mid-range of density. This is a long-standing theoretical problem, but other sources of uncertainty make it difficult to highlight specifically this problem in the measurements.

- In the study by TK21, they do not refer to Strong Contrast Expansion (SCE). While the authors' descriptor is useful, SCE in our community generally refers to snow cover extent which might be confusing for some readers. By way of a suggestion, perhaps the authors could use an alternative descriptor?

Rechtsman and Torquato, 2008, Kim and Torquato 2000 and TK21 use "strong-contrast expansion" in many places. Although we understand Reviewer's concern, we feel uncomfortable with renaming their theory and we still need an acronym.

- Line 35. In the paper (TK21), they did not specify a snow medium but a 2-phase medium more generally. The authors make it seem as though TK21 had derived it explicitly for snow. Perhaps re-phrase "….which can be used to express the dielectric polarizability…."

TK21 is generic while this paper is a specific application to snow. It was not explicit in our first submission. We have reforumated this section to make it clear, and in particular:
" with respect to the background material as an \emph{exact} infinite series in the dielectric polarizability of the foreground material with respect to the background. Here, we consider the particular case of snow, assumed to be an isotropic 3-dimensional medium composed of ice (foreground) and air (background)."

- Equation 1, and Line 47 –  As is not defined.  It is sort of defined on Lines 51-52 but still not very clear. Please can the authors define this term as it is in equation 1.

We have added a forward reference to the paragraph dedicated to these terms.

- Line 41 - Is this the same as the scaled SCE in Figure 1? If so, be sure to use consistent nomenclature. If not, then can the authors explain the difference more clearly?

The scaled SCE is defined in TK21 and we use the scaled SymSCE in Figure 1 by applying symmetrization to the scaled SCE following TK21 approach described in their Appendix. We are sure that the nomenclature is correct and consistent.
However, we have shifted the quote which may have been a source of confusion: "scaled" SCE in TK21→ "scaled SCE" in TK21

- Line 46 – it is not clear what the italicized symbol is in the in-text definition of Ke. This should be defined.

This is the Latex symbol for the imaginary part. We have added the definition.

- Line 57 - what, specifically, would be needed in terms of 'increasingly more detailed information of microstructure?' What does this mean? The authors should provide the reader with more clarity since field experimentalists will be interested.

We propose to add "(e.g. larger and more resolved 3D images of the snow microstructure)". Although this statement clarifies the text for the "experimentalists", it suffers from subjectivity, mainly because it is not yet clear how the size and resolution of microstructure images translate into accuracy of the n-point correlation functions and then in the An terms.

- Lines 56 – 59 – What is the benefit of n > 2 should it be a feasible computation? Or, in other words, what are we missing out on by using n = 2? What are the practical implications?

As in the previous comment, we do not know how different the model would be with n=3. It depends on the amplitude of A3 and of the 3-point correlation function.

Our feeling is that given the uncertainties in fitting the 2-point correlation functions models to experimental snow data (see Sandells et al. 2021) , we believe the priority in 2022 is on consolidating the case n=2 before addressing n=3 in the future, but the latter remains an interesting piece of work. In both cases anyway, this represents very significant work for our community as detailed in our response to Reviewer 1's comments. It is very far beyond the scope of this paper.

- Figure 1 and lines 70 – 71, Line 80 (and discussion throughout paper) – RT08 does not appear in Figure 1. Do the authors mean SCE R08? Similarly, TK21 in Figure 1. Do you mean Scaled SymSCE T21?  The authors should check that the text matches the figure descriptors, here, and throughout the paper.  Otherwise, it is confusing.

We have corrected the typo on RT08 and TK21 in Figure 1.

- Line 78 – what is the FDTD method ? Please expand this acronym.

Done

- Line 80 – The authors imply that there is some spread between Scaled SymSCE T21 and Mie DMRT and IBA for koa>~1. What are the implications, or is this unimportant?

We interpret the equivalence of the theories below <1.5 as a good sign (but not as a proof). The differences between the theories is suspicious and in any case at most one can be correct. For this reason, we conclude that it is safer to keep the size < this limit when using any of these theories, but this is not a demonstration that these theories are valid for up to 1.5.
We have reformulated the text: "These results also highlight the equivalence between SCE, IBA and DMRT Mie up to about $k_0 a \sim 1.5$. Based on these results, our main practical recommendation is limiting the size in the range where these three theories agree."

- Line 122 – the authors write "Let choose a linear combination…"  This should be "Let us choose…"

Done

- Line 125 – you introduce SymSCE. Is this the same as Scaled SymSCE T21 shown in Figure 1? This gets a little confusing.

We have corrected this issue by adding scaled: " This provides a new approximation of the scaled SCE that we call scaled SymSCE."

- Line 149 – 151 - Is there work going forward on this? There must be some existing suite of in situ measurements that could satisfy the requirements, no?

Not to our knowledge. A relatively homogeneous snowpack around 450-500kg/m3 is not frequent, and assessing these theories over thick, and thus heterogeneous, snowpacks (as on the ice-sheet) is subject to many other uncertainties.

- Line 155 – the authors call for more precise in-situ microwave and snow measurements than what is available at present. What do you mean by more precise? Please be more specific here – what measurements, exactly, are needed? How does 'SCE' relate to physical field measurements? What field measurements are needed?

We have added the following information

"will require more precise in-situ active or passive microwave (backscatter or brightness temperature) and snow measurements (density, grain size, temperature) from snowpack with intermediate densities (450 – 500 kg m$^{-3}$) than available at present."

---

## Referee Report (RR1)

The authors have responded adequately to my previous review.

I recommend the following minor revisions

The authors should state the following in the Introduction or Conclusion

>The order of correlation function for the iteration approach has been limited to n=2 in this paper. The order n=3 is complicated and has not been studied. Did Torquato's group perform calculations beyond n=2? Please comment

---

## Author Response (AR2)

Response to reviewer #1.

"The authors should state the following in the Introduction or Conclusion
The order of correlation function for the iteration approach has been
limited to n=2 in this paper. The order n=3 is complicated and has not
been studied. Did Torquato's group perform calculations beyond n=2?
Please comment ?"

The information about the limitation n=2 is in Section 2, in Section 3,
and in Section 6 (the conclusion). To make it more explicit we have
amended (bold face) the conclusion as follows:

"Nevertheless, the SCE as presented in TK21 is even more general than
applied here **(isotropic medium and truncation at n=2)**, paving the
way to further improvements. For instance, the calculations of the higher
terms of the series (n>2) **is complicated but** should extend the validity
range to higher frequencies and/or coarser-grained snow, provided that
more detailed microstructure information could be obtained …"

TK21 does present the theoretical equations with all the orders n. It also
derives specific equations for n=2 only, and addresses some numerical
issues for n=2 –  which proved to be crucial for our numerical
implementation – but not beyond (n>2). This is why further
mathematical / numerical work is required to apply this theory for n>2.
In addition, collecting the relevant information on microstructure in the
case of snow is a specific problem, as discussed in the aforementioned
sections.